# Dimeric (Poly)Hydroxynaphthazarins, Metabolites of Echinoderms and Lichens: The History of the Synthesis and Structure Elucidation

**DOI:** 10.3390/md21070407

**Published:** 2023-07-19

**Authors:** Dmitry N. Pelageev, Ksenia L. Borisova, Victor Ph. Anufriev

**Affiliations:** G. B. Elyakov Pacific Institute of Bioorganic Chemistry, Prospect 100 let Vladivostoku 159, 690022 Vladivostok, Russia; pelageev@mail.ru (D.N.P.); borisovaksenia@mail.ru (K.L.B.)

**Keywords:** ethylidene-bis(2,6,7-trihydroxynaphthazarin), octahydroxymethyldibenzo[*b,i*] xanthenetetraone, mirabiquinone, cuculoquinone, islandoquinone, hybocarpone, mesocentroquinone, sea urchins, lichens, metabolites, tautomerism

## Abstract

This review provides information on the synthesis and revision of the structures of natural dimeric (poly)hydroxynaphthazarins, metabolites of echinoderms and lichens, and on the refinement of the direction and mechanism of reactions in the synthesis of some of these compounds.

## 1. Introduction

Compounds based on the structure of 1,4-naphthoquinone are widespread in natural objects [1,2,3,4,5,6,7,8,9]. Among thousands of its derivatives, including dimeric and trimeric products [1,2,3,10,11,12], derivatives of naphthazarin (5,8-dihydroxy-1,4-naphthoquinone) occupy a very prominent place [13,14,15,16,17]. Among them, dimeric (poly)hydroxynaphthazarin metabolites of echinoderms and lichens constitute a relatively small but structurally diverse group of biologically active natural compounds. The first representatives of this series of compounds were isolated in the early 1970s, yet little information on the isolation of new dimeric products of this group, their synthesis, and correction of the structures of previously isolated compounds appears in print to this day. Our research initially concerned the synthesis of compounds of this group, with the aim of obtaining them in quantities required for biotests. However, it soon became clear that the real structures of some of the compounds did not correspond to the declared ones. This is primarily due to the imperfection of the physicochemical equipment that existed at the time of the study and the lack of or incorrect interpretation of the information available. It should be noted that ignorance of the exact structures of substances makes it impossible to create structure-activity correlations and, therefore, a targeted search for substances with desired properties. In this situation, synthesis, in addition to a supplier of substances with a given structure, plays the role of a reliable tool for its analysis. This review provides information on the synthesis of natural dimeric (poly)hydroxynaphthazarins, metabolites of echinoderms and lichens, as well as some information on the revision of their structure and the mechanism of formation. In this way, this review differs from the recently published [18,19], in which significant attention was paid to the isolation and structures of spinochromes, as well as the assessment of their biological activity, the parent organisms, and the methods used for isolation and identification. In addition, attention was paid to the study of the biosynthesis of spinochromes and the ecological function, stability, and chemical synthesis of (poly)hydroxynaphthazarins.

When analyzing the structures of the above compounds, we can conclude that all, formally, are the end products of three types of reactions. The first subgroup consists of dimeric (poly)hydroxynaphthazarins, which are the end products of the aldol condensation reaction. The initial 2-hydroxynaphthazarins act as methylene components in this reaction. The second part of the review consists of dimeric (poly)hydroxynaphthazarins, products of oxidative C-C or C-O dimerization. Finally, the third subgroup includes the only one synthesized to date and, then, dimeric (poly)hydroxynaphthazarin, recently discovered in sea urchins, a product of heterodiene condensation, in which diene and dienophile are two forms of the same hydroxynaphthazarin derivative.

## 2. Types of Dimeric (Poly)Hydroxynaphthazarins

### 2.1. Aldol Condenation Compounds

The first report on the isolation of this type of product appeared in 1971, when ethylidene-bis(trihydroxynaphthazarin) **1** (Figure 1) was isolated from an extract of the sea urchin *Spatangus purpureus* (Figure 2) [20]. The structure of compound **1** was reliably established using UV, IR, and ^1^H NMR spectroscopy and mass spectrometry. In addition to product **1**, another product was isolated from the extract, the structure of which, due to its small amount, could not be reliably determined. In the ^1^H NMR spectrum of this product, as in the spectrum of **1**, proton signals of the ethylidene bridge connecting the hydroxynaphthazarin fragments were observed, and the mass spectrum showed a peak of the molecular ion with *m*/*z* 484, eighteen mass units less than the peak of the molecular ion of compound **1** (*m*/*z* 502). This suggested that the product is an anhydro derivative of ethylidene-bis(trihydroxynaphthazarin). In order to test this assumption, an attempt was made to synthesize this compound from substrate **1**. Thus, heating **1** in concentrated sulfuric acid gave a product in low yield which, according to the UV, mass spectrum, and R_f_ value, is an anhydro derivative of ethylidene-bis(trihydroxynaphthazarin). This product can correspond to one of the three isomeric dibenzo[*b,i*]- (**2**), [*b,h*]- (**3**), and [*c,h*]- (**4**) xanthetetraones (Figure 1). At the same time, in the IR spectrum of the obtained product (KBr) there are two absorption bands of carbonyl at 1621 and 1600 cm^−1^, while in the spectrum of the natural product, there is only one broad multi-shouldered carbonyl band at 1600 cm^−1^. Despite this, the authors, having analyzed all the data at their disposal, made a choice in favor of structure **2** [20]. At the same time, they indicated that the final conclusion about the structure of the cyclization product can be made by the number of methoxy groups (tetra-**2**, penta-**3**, hexa-**4**) obtained by methylation of *β*-hydroxy groups of compound with diazomethane (^1^H NMR).

Thus, the structure of the dehydrated ethylidene-bis(trihydroxy-naphthazarin) derivative **1** remained not fully elucidated, and therefore it is not mentioned in the fundamental monography by Thomson, R. H. [2]. At the same time, some authors referred to it as proven [21], and only in 2017 was complete spectral information on the structure of dibenzo[*b,i*]xanthetetraone **2** [22].

Recently, mirabiquinone (1*H*-dibenzo[*b,h*]xanthenetetraone, **3**) was isolated from the sea urchin *Scaphechinus mirabilis* (Figure 2) [23], which was previously considered as an alternative to 5*H*-dibenzo[*b,i*]xanthenetetraone **2** isolated from the sea urchin *S. purpureus* [20]. In the IR spectrum of mirabiquinone (CHCl_3_), there is one absorption band of carbonyl at 1626 cm^−1^. Comparison of the IR spectra of the anhydro derivative **2**, mirabiquinone (**3**), and the cyclization product of ethylidene bis(trihydroxynaphthazarin) **1**, under the action of concentrated sulfuric acid [20], made it possible to establish that the latter Is a mixture of dibenzo[*b,i*]xanthetetraones **2** and **3**.

Mirabiquinone (**3**) and two related binaphthazarins **1** and **2** demonstrated excellent scavenging of the 2,2-diphenyl-1-picrylhydrazyl radical [23]. However, these compounds are not easily accessible on a preparative scale for extended bioassays due to their very low natural abundance and separation difficulties. Therefore, the question of the synthesis of these compounds for biotesting has become pertinent.

The key stage in the synthesis of biquinone **1** was the aldol condensation of spinochrome D dimethyl ether **5** and acetaldehyde (Figure 1) [24]. Demethylation of tetramethyl ether **6a** by the action of AlCl_3_ in nitrobenzene gave ethylidene-bis(thrihydroxynaphthazarin) **1** in good yield [25]. An attempt of demethylation of tetramethyl ester **6a** by the action of conc. HBr yielded 5H-dibenzo[*b,i*]xanthetetraone **2**, previously isolated from the sea urchins *S. purpureus* [20] and *S. droebachiensis* [21], and mirabiquinone A (**3**), a metabolite of the sea urchin *Scaphechinus mirabilis* [23], in a ratio of 1.2:1.

It is obvious that the cyclization of ethylidene-bis(thrihydroxynaphthazarin) **1** and its derivatives is a key step for the preparation of 5*H*-dibenzo[*b,i*]- (**2**) and 1*H*-dibenzo[*b,h*]- (**3**) xanthenetetraone. It has been found that the boiling of tetra- (**6a**), penta- (**6b**) or hexamethyl (**6c**) ethers in toluene in the presence of p-TsOH gave the corresponding anhydro derivatives **7a** and **7b** in good yields (Figure 2) [26]. The generation of mirabiquinone (**3**) from its tetra- (**7a**) and penta- (**7b**) methoxy derivatives was affected by exposure to AlCl_3_-EtSH in CH_2_Cl_2_.

As suggested by the authors of [20,21], the cyclization of ethylidene-bis(trihydroxynaphthazarin) **1** in concentrated sulfuric acid produces anhydrous derivative **2**. To obtain some amount of this substance for biological testing, a replica of this experiment was conducted. Surprisingly, it was found that, under the described conditions, ethylidene-bis(trihydroxynaphthazarin) **1** undergoes cyclization with the formation of mirabiquinone (**3**) instead of 5*H*-dibenzo[*b,i*] xanthenetraone **2** in good yield [26]. This suggests that UV-, IR-spectroscopy, and TLC are not reliable enough at establishing structures of polyhydroxynaphthazarins, even by the comparison method. For example, these methods cannot distinguish between bisnaphthazarin **1** and related naphthazarins **2** and **3,** which were considered in the cited work [20].

### 2.2. Oxidative Coupling Compounds

Other representatives of this group of compounds are dimeric (poly)hydroxynaphthazarins **8–10** [27,28,29], which are products of the oxidative C-C or C-O coupling of ethylmompain (**11a**) and boryquinone (**11b**) (Figure 3).

As noted, natural dimeric (poly)hydroxynaphthazarins are often available only in small amounts, which hinders the use of chemical methods for the establishment of their structures, whereas the available physicochemical methods do not allow unambiguous conclusions about the arrangement of the substituents in the quinoid moiety at the C(2) and C(3) atoms with respect to the substituents at the C(6) and C(7) atoms. This fully applies to cuculoquinone, hydroxylated bisnaphthazarin isolated from the red thallus tips of the lichen *Cetraria cucullata* [27]. Cuculoquinone is one of the three identifiable quinonoid compounds produced by this lichen species, which grows in the Magadan region of Russia. For this compound, the following structure, 3,3′-bis(7-ethyl-1,4,5,8-tetrahydroxy-2,6-naphthoquinone) (**12**) (Figure 3), was proposed. The *amphi*-(2,6)-quinonoid structure of bisnaphthazarin **12** raised doubts in Thomson R. H., who proposed, for this product, the structure of 3,3′-bis(7-ethyl-2,5,6,8-tetrahydroxy-1,4-naphthoquinone) (**13**), i.e., 1,4-naphthoquinonoid structure, in which the *β*-hydroxy groups of each fragment are located in positions 2 and 6 [2].

Later, a series of substituted 2,6- and 2,7-dihydroxynaphthazarins were synthesized. It is shown that the absorption bands in the ultraviolet and visible regions of the electronic spectrum of alkaline solutions, as well as the frequencies of stretching vibrations of *β*-O-H in the IR-spectrum of 2,6- and 2,7-dihydroxynaphthazarins have characteristic, non-overlapping ranges of values [30]. The regularities found made it possible to revise the structure of cuculoquinone, isolated from the lichen *C. cucullata*, into 3,3′-bis(6-ethyl-2,5,7,8-tetrahydroxy-1,4-naphthoquinone) (**8**).

In addition, compounds **13** and **8** were synthesized, and the latter was shown to be completely identical to bisnaphthazarin isolated from the lichen *C. cucullata* [31]. Thus, treatment of monomethyl ether **14** with (NH_4_)_2_S_2_O_8_ in MeCN-H_2_O gave the bisnaphthazarin dimethyl ether **15** (30%) (Figure 4). Compound **15** was easily converted into the corresponding bis(2,6-dihydroxynaphthazarin) **13** by the action of AlCl_3_ in nitrobenzene (44%). In the same way the ether **16** via dimethyl ether **17** was converted into bis(2,7-dihydroxynaphthazarin) **8** (total yield 21%).

Bisnaphthazarin **8** has been found in the deep-sea holothuroids *Psychopotes longicauda*, *Benthodytes typica*, *B. lingua* [2], and in the lichen *C. islandica* [28] (Figure 5). The well-studied mechanism of biosynthesis of compounds analogous to spinochromes [32,33] can be considered as circumstantial evidence in favor of structure **8**. The structures of all natural naphthoquinone derivatives containing the 2,7 dihydroxynaphthazarin fragment as a subgroup [2,29,34,35,36,37] are consistent with the above mechanism.

Another formal product of oxidative dimerization is islandoquinone, a metabolite isolated from the lichen *C. islandica* [28]. Structure **18** was proposed for this compound as a result of the comparison of the compound obtained with the lapachol peroxide structure **19** (Figure 6) [38,39]. However, the IR-spectrum of islandoquinone did not contain *ν*_(C=O)_ absorption bands at ≈1750 cm^−1^ that are found for 2,3-dihydro-2-oxo-1,4-naphthoquinones [40].

Accordingly, the 2,3-dihydro-2-oxo-1,4-naphthoquinonoid structure of the Q_2H_ fragment of biquinone **18** was revised, and this natural product was identified as **20**, i.e., the 2,3-dihydro-2,2-dihydroxy-1,4-naphthoquinonoid structure was assigned to the Q_2H_ subgroup [41]. An argument in favor of structure **20** was based on the comparison of its spectral data with those of the 2,3-dihydro-2,2-dihydroxy-1,4-naphthoquinones described in the literature [42,43]. However, doubts have emerged regarding the proposed structure of islandoquinone, with the major discrepancy concerning the presence of proton signals of only three *α*-hydroxy groups in the ^1^H NMR spectrum of islandoquinone [28].

Based on the accumulated spectral data, it was concluded that islandoquinone is one of four dioxabenzo[*a*]tetracenetetraones from the two diastereoisomeric pairs of **9**, **9′** and **21**, **21′** (Figure 7). According to quantum chemical calculations [44], the diastereoisomers 7a*S**,13a*S**- (**9**) and 7a*R**,13a*R**- (**21**) are more favorable than the corresponding diastereoisomers 7a*S**,13a*R**- (**9′**) and 7a*R**,13a*S**- (**21′**). The difference in Gibbs energy between **9** and **21** is only 0.4 kcal/mol [45]. The conclusive choice in favor of either **9** or **21** may be based on the X-ray diffraction analysis of islandoquinone or structurally similar compounds.

Within this context, the oxidative coupling products of ethylhydroxynaphthazarins **22a** and **22b** (Figure 8) [42] were synthesized and analyzed. The oxidative coupling of the chlorinated hydroxynaphthazarin **22a** upon treatment with lead dioxide in boiling acetic acid resulted in a product that, judging from the spectral data, was an unsymmetrical biquinone [45]. In the case of cristazarin (**22b**), a mixture of two biquinones (1:1.8 ratio, ^1^H NMR) was produced. The crystallization of the chlorinated biquinone and major product of oxidative dimerization of cristazarin (**22b**) from acetone afforded crystals that were suitable for single-crystal X-ray diffraction.

The molecular structures of the obtained products and their corresponding structural formulas (**23a**,**b**) are shown in Figure 9 [45].

The upfield signals of the ethyl group protons at C(15), C(16), C(17), and C(18) and carbon atoms C(6a), C(7a), C(13a), and C(14a) (Table 1) of the dioxane ring of the biquinones **23a,b** were in very good agreement with the corresponding signals of islandoquinone [28,42]. Therefore, the connection of rings B and C and the position of the substituents in these rings in these biquinones and islandoquinone are identical. These data indicate that the early proposed structure of islandoquinone should be revised in favor of dioxabenzo[a]tetracenetetraone **9** and compound **23b** is its dimethyl ether. All attempts to convert **23b** into islandoquinone were unsuccessful at yielding a complex mixture of compounds.

On the other hand, based on the B3LYP/6-311G(d) method [44], it was concluded that the minor product obtained by oxidative dimerization of cristazarin is dioxabenzo[a]tetracenetraone **23c** (Figure 9). In the ^1^H NMR spectrum of compound **23c**, the upfield signals of the ethyl group protons at C(15), C(16), and C(18) and carbon atoms C(7a) and C(13a) do not fit with those of islandoquinone and dioxabenzo[a]tetracenetetraones **23a,b** (bolded in Table 1). Therefore, these signals are important structural evidence.

The cytotoxic pentacyclic naphthazarin-derived dimer, hybocarpone (**10**), was isolated from the lichen *Lecanora hybocarpa* by Elix J. A. and co-workers in 1999 [29] (Figure 10).

Several years later, the synthesis of **10** and the related (5aS*,6aS*,12aS*,12bS*)-binaphtho[2,3-b; 2,3-d]furantetraones, **24a,b** (Figure 11) was realized [46,47,48].

The total synthesis of hybocarpone involves two key synthetic steps: the formation of 2-hydroxy-1,4-naphthoquinone **25a** from substituted benzaldehyde **26** and its oxidative dimerization on treatment with CAN in MeCN (Figure 4) [46,47].

More recently, another route to substituted 1,4-naphthoquinone **25a** from flaviolin trimethyl ether **28** [49,50], *o*-naphthoquinone **29** [51], and *α*-naphthol **30** [52] were proposed (Figure 12). These approaches include the multistep synthesis of both the key substrates **26**, **28–30** themselves, and their subsequent conversion to hybocarpone (**10**).

By analogy with hybocarpone, the related binaphtho[2,3-*b*; 2,3-*d*]furantetraones **24a**,**b** were synthesized from the corresponding derivatives of 1,4-naphthoquinone **25b**,**c** [47] (Figure 13).

In another synthetic approach to hybocarpone (**10**) and its analogs, the use of the direct oxidative dimerization of the 2-hydroxynaphthazarin precursors **31a**,**b** was explored (Figure 14) [53]. These compounds are more readily available [54,55] than the 1,4-naphthoquinone precursors **25a–c**. However, all attempts to construct the appropriate binaphtho[2,3-*b*; 2,3-*d*]furantetraone skeleton by the action of CAN in MeCN [47] led to degradation of the starting structures **31a**,**b**. Upon screening a number of reagents and conditions, success was finally achieved with the use of Pb(Oac)_4_ as an oxidant in benzene.

Oxidative coupling of hydroxynaphthazarin (**31a**) on treatment with Pb(Oac)_4_ in benzene gave two compounds in a ratio of ca. 1:1. One of them exhibited a simple ^1^H NMR spectrum reminiscent of that of **10**, and judging from the spectral data, was binaphtho[2,3-*b*; 2,3-*d*]furantetraone (**32a**) (Figure 14). As such, the upfield ^1^H NMR signals of the ethyl group protons and ^13^C NMR signals of carbon atoms of the tetrahydrofuran ring of **32a** were in very good agreement with the corresponding signals of hybocarpone (**10**) [29].

The other isomeric biquinone exhibits the correct mass (by mass spectrometry) and simple ^1^H and ^13^C NMR spectra, but the upfield signals of the ethyl group protons and carbon atoms of the tetrahydrofuran ring of that compound do not fit those of **10**. The crystallization of this biquinone from hexane-acetone afforded crystals that were suitable for single-crystal X-ray diffraction. The molecular structure of the product obtained (**33a**) is shown in Figure 15.

Thus, the oxidative coupling of hydroxynaphthazarin (**31a**), on treatment with Pb(Oac)_4_ in benzene, gave the diastereomeric mixture of 5a*S**,6a*S**,12a*S**,12b*S** (**32a**) and 5a*S**,6a*R**,12a*R**,12b*S** (**33a**), analogs of hybocarpone (**10**). The ratio of the arising compounds **32a** and **33a** (1:1) was determined by the equally possible formation of the intermediary *S***S** (**34a**) and *R***S** (**35a**) diastereomers (Figure 5). So, according to the quantum chemical calculation [44], the difference between Gibbs energy of diastereomers **32a** and **33a** is less than 0.4 kcal/mol) [53].

The hydration/cyclization of intermediary *S***S** (**34a**) and *R***S** (**35a**) diastereomers would potentially lead to the formation of up to six diastereomeric furan systems arranged in two rows of three (Figure 16). Molecular modeling and computational studies indicated that, among the diastereoisomeric compounds in each row, the isomers **32a** and **33a** appeared to be clearly favored in terms of relative Gibbs energy [44,56]. Since the calculated energy differences among compounds in each row are large (more than 9 kcal/mol), and because the central dihydroxyfuran systems of them can exist in equilibrium with their open chain counterparts, diastereoisomers **32a** and **33a** are the only imaginable products in this reaction.

These observations were used as the basis for the synthesis of hybocarpone (**10**). Methylcristazarin (**31b**) is a more available substrate for this purpose. As in the case of **31a**, the oxidative coupling of hydroxynaphthazarin (**31b**), on treatment with Pb(Oac)_4_ in benzene, gave two products. One of them, according to the spectral data, was hybocarpone dimethyl ether **32b** (Figure 14). The other product was the isomer **33b** bearing a *sin* relationship of the two ethyl groups at the junction joining the two monomeric units.

Dimethyl ethers **32b** and **33b** were deprotected with AlCl_3_ in EtSH-CH_2_Cl_2_ to afford **10** and **36** (Figure 17). Synthetic **10** exhibited spectral data (^1^H and ^13^C NMR, mass spectrometry) identical to those reported for natural hybocarpone, a cytotoxic metabolite isolated from *L. hybocarpa* lichen [29].

It should be noted that in the previous report [47], the course of the reaction through the *S**,*S** hexaketone intermediate **27** (Figure 4) was only postulated, and the possibility of *R**,*S** diastereomer formation was not discussed; thus, we compared our results for the oxidative dimerization of dihydrolapachole **25c** with Pb(Oac)_4_ in benzene [57] to those previously reported using CAN in MeCN [47].

Oxidative dimerization of dihydrolapachole **25c** upon treatment with Pb(OAc)_4_ in benzene yielded three products following chromatographic purification [57]. A yellow product established as **24b** by Nicolaou [47]was determined to be 3-(naphthoquinone-2-yloxy)naphthalenetrione **37** (42%) based on its spectral data (^1^H, ^13^C NMR and mass spectrometry) and comparison with the authentic sample [58] (Figure 18). The formation of a high percentage of **37** resulted due to the steric bulk around the reacting carbon in starting substrate **25c**; thus, C-O coupling was more probable than C-C coupling.

Two colorless products were determined to be pentacyclic compounds **24b** (24%) and **24c** (21%) based on the spectral data and comparison of its ^1^H and ^13^C NMR data with spectral data of derivatives **32a**,**b** and **33a**,**b**. We found that the same mixture was obtained upon exposure of the monomeric unit **25c** to CAN in MeCN under the previously reported conditions [47].

In general, 2-hydroxy-3-alkylnaphthazarins and 2-hydroxy-3-alkyl-1,4-naphthoquinones undergo oxidative dimerization upon treatment with lead tetraacetate or cerium ammonium nitrate in aprotic media to give diastereomeric 5a*S**,6a*S**,12a*S**,12b*S**- and 5a*S**,6a*R**,12a*R**,12b*S**-dihydrobinaphthofurantetraones in a ratio of ca. 1:1. The ratio of arising compounds is determined by the equally possible formation of the corresponding *S**,*S** and *R**,*S** hexaketone intermediates.

### 2.3. Diene Condenstion Compounds

Among natural partially methylated derivatives of echinochrome, there are no examples of structures that simultaneously contain 2-hydroxy and 3-(1-hydroxyethyl) groups in one core [1,2,3]. This is obviously due to the instability of such compounds. Indeed, attempts to synthesize 2-hydroxy-3-(1-hydroxyethyl) naphthazarin **38** by alkaline hydrolysis of lomazarin (**39a**), its 1′-bromo- (**39b**) and 1′-acetoxy- (**39c**) derivatives are invariably resulted in spinochrome D dimethyl ether **5** (Figure 19). Most likely, 1′,2-dihydroxy-3-ethylnaphthazarin **38**, formed from starting compounds **39a**–**c** under basic conditions, is converted to dimethyl ether **5** via the mechanism of retroaldol decomposition of intermediate keto form **40** [24,25].

Our attempts to synthesize compound **38** by bromination of echinochrome dimethyl ether **41** and subsequent hydrolysis of 1′-bromo derivative **42** already at the first stage led to an unexpected result. The final and main product of the reaction was 2-naphthoquinonylbenzo[*g*]chromendione **43** (up to 80%) (Figure 6) [59].

The formation of benzo[*g*]chromene-5,10-dione **43** likely occurs via the mechanism of heterodiene condensation (Figure 6). The 1′-bromoethyl derivative **42** formed during the reaction loses HBr, giving the enone **44** (heterodiene), which are isomerized to the corresponding vinylquinone **45** (dienophile). The isomerization of **44**→**45** is reversible, since at the end of the reaction the starting diene and dienophile are not found in the mixture. It should be noted that 1′-bromoethyl derivative **42** are labile compounds. In acidified solutions of chloroform, acetone, or on the surface of H^+^-silica gel, they are rapidly converted into the corresponding benzo[*g*]chromene-5,10-dione **43**. The structure of product **43** and stereochemistry of its asymmetric centers are unambiguously determined by analysis of their ^1^H and ^13^C NMR spectra [59]. Hydrolysis of benzo[*g*]chromedione **43** gave product **46** (Figure 20).

When this work was in progress, a previously unknown pigment was detected in the extracts of the sea urchins *Mesocentrotus nudus* and *Strongylocentrotus intermedius* by HPLC-MS method (Figure 21) [60]. The retention time, UV, and mass spectra of the detected product coincided with those of synthesized compound **46** [59].

Thus, the discovered product, which was named mesocentroquinone, has the structure 6,7,8,9-tetrahydroxy-4-methyl-2-(3,5,6,7,8-pentahydroxy-1,4-dioxo-1,4-dihydronaphthalen-2-yl)-3,4-dihydro-2*H*-benzo[*g*]chromene-5,10-dione. In fact, it can be considered as a dimer of dehydro derivative of echinochrome (**47**) obtained by diene condensation.

## 3. The Tautomerism of Hydroxynaphthazarins

NMR spectroscopy is among the most used methods for the structural study of hydroxynaphthazarins [1,2,3]. The phenomenon of tautomerism inherent in the naphthazarin system leaves an imprint on the nature of the spectra of substituted naphthazarins, including its hydroxy derivatives [61]. Due to the rapid (on the NMR time scale) tautomerism, in the spectra of naphthazarin and its derivatives, the signals of protons and carbon atoms entering the quinoid and benzenoid cycles are indistinguishable in pairs. Thus, in the ^1^H NMR spectrum of mompain monomethyl ether **48** (CDCl_3_), the signals of protons adjacent to the hydroxy and methoxy groups of tautomers Q and B are indistinguishable (Figure 22) [54]. Thus, on the NMR scale, mompain monomethyl ether is an individual compound.

IR spectroscopy is a much faster method when compared to nuclear magnetic resonance spectroscopy, in which there is usually no temporal averaging of spectral parameters. Therefore, the IR spectrum of mompain monomethyl ether **48** taken in CDCl_3_ showed that this compound is a mixture of 1,4-naphthoquinoid tautomers **48**(Q) and **48**(B), and in commensurate proportions (70% and 30%, respectively) [62,63]. In addition, quantum chemical calculations, using the example of 1′-hydroxyalkyl naphthazarin **6** [64], showed that the energy barrier for a process of type **49**(Q_1.4_) ⇆ **49**(Q_1.5_) (Figure 22) is less than 5 kcal/mol [64], which makes possible the existence of corresponding 1,5-naphthoquinoid forms. Later, it was shown by IR spectroscopy that in aprotic organic solvents, 1′-hydroxyalkyl naphthazarins are in the form of a mixture of 1,4- and 1,5-naphthoquinoid tautomers [65].

Thus, due to the easily reversible conversion of specified tautomers, it is impossible to isolate any component of the mixture in an individual form. This is also true for dimeric (poly)hydroxynaphthazarins, which are in no way different from monomers in this respect. At the same time, recent studies have reported the isolation of several tautomers in individual form by the HPLC method and the establishment of their structure by NMR spectroscopy [18,19,66]. The most likely reason for this misunderstanding was the ability of (poly)hydroxynaphthazarins to give stable crystal solvates and chelate-type derivatives, which ultimately led to erroneous conclusions about the structure of the isolated products. This misconception has a long history, the beginning of which lies in the first messages about the allocation of (poly)hydroxynaphthazarins from natural objects [67,68,69].

## 4. Conclusions

This review provides information on the establishment of the structure of natural dimeric (poly)hydroxynaphthazarins, metabolites of echinoderms and lichens. Due to the relatively low content of these products in natural objects, and for several other reasons, the establishment of their structure has encountered certain difficulties. Success in overcoming this issue was achieved by using modern physico-chemical research methods and counter synthesis. The results of these studies were the revisions of the structures and syntheses of metabolites of lichens *Cetraria ‘olothuri*, *C. islandica*, deep-sea holothuria *Psychropotes longicauda*, and a representative of the genus Benthodytes. The structure of islandoquinone, a metabolite of the lichen *C. islandica*, the backbone of which is dioxabenzo[*a*]tetracenetetraone, underwent a serious correction. Mesocentroquinone, the structure of which is based on benzo[*g*]chromedione, was synthesized earlier than it was isolated from the sea urchins *Mesocentrotus nudus* and *Strongylocentrotus intermedius*. In addition, the review provides information on clarifying the direction and mechanism of reactions in the synthesis of some natural dimeric (poly)hydroxynaphthazarins. This refers to the conversion of ethylidene-bis(trihydroxynaphthazarin) to linear dibenzo[*b,i*]xanthenetetraone, both of which are metabolites of the sea urchin *Spatangus purpureus*. Relatively recently, it was found that, as a result of this reaction, the angular dibenzo[*b,h*]xanthenetetraone, mirabiquinone was also formed, which was isolated from the sea urchin *Scaphechinus mirabilis* and synthesized later. Another example is also the clarification of the mechanism of the key synthesis reaction of hybocarpone, a metabolite of the lichen *Lecanora hybocarpa*.

## Data Availability

Data sharing not applicable.

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
