# Peer review of "Dimeric (Poly)Hydroxynaphthazarins, Metabolites of Echinoderms and Lichens: The History of the Synthesis and Structure Elucidation"

_marinedrugs, 2023, doi:10.3390/md21070407_

Round 1

Reviewer 1 Report

The review article by Anufriev et al focuses on the synthesis and structure of dimeric (poly)hydroxynaphthazarins, a little-known class of echinoderm and lichen metabolites possessing radical scavenging and cytotoxic activities. The emphasis is on the correct structural assignment of these natural products which includes numerous significant contributions from the author’s laboratory.

Indeed this article highlights the challenges associated with the structural elucidation of the tittle compounds and should be of interest to a number of natural product chemists.

The manuscript is written with clarity and contains few errors. I therefore support its publication in Marine Drugs after minor revision.

Minor Error (Conclusions, line 450):  Change “supplier” to “availability”.

Author Response

We are grateful to the reviewer for reviewing our manuscript.

Minor Error (Conclusions, line 450):  Change “supplier” to “availability”.

The remark is irrelevant, since the conclusions were completely revised on the recommendation of another reviewer.

Reviewer 2 Report

In this manuscript, Victor Ph. Anufriev and coworkers gave us an insight to the history of the structure elucidation and revisions via chemical synthesis. Overall, this work is interesting and important.

However, there are some concerns as followings:

1. Please add the Gibbs energy for compounds 9 and 21 in Figure 7, for 32a and 33a in Scheme 5, for 49(Q1.4) and 49(Q1.5) in Figure 22.

2. For the convenience of readers' understanding, it is better to add the notations δH and δC in Table 2.

3. As many diastereomers were found and synthesized, were there reported absolute configurations of any compounds?

4. The paragraph in the subsection Conclusions was the same as part of the first paragraph in the subsection Introduction. Please rewrite it. You can list the correct structures of some compounds in this part.

5. Please update the information for Reference [58].

Minor editing of English language required. Only a few typo or grammar errors, such as ‘...raised doubts about Thomson R. H.’ (P5L150), ‘... the reaction lose HBr’ (P14L385), ‘... compound 46 synthesized’ (P14L399), including the Italics for the Latin names such as ‘Scaphechinus mirabilis’ (P3L90), ‘S. purpureus’ (P3L91), ‘S. droebachiensis’ (P3L107), and the compound names such as ‘dibenzo[b,i]xanthetetraone’ (P3L85), ‘1H-dibenzo[b,h]xanthenetetraone’ (P3L89), ‘5H-dibenzo[b,i]xanthenetetraone’ (P3L90).

Author Response

We are grateful to the reviewer for reviewing our manuscript.

Comments and Suggestions for Authors

  1. Please add the Gibbs energy for compounds 9and 21in Figure 7, for 32a and 33a in Scheme 5, for 49(Q1.4) and 49(Q1.5) in Figure 22.

The difference of Gibbs energies for compounds 9 and 21 (Fig. 7), 32a and 33a (Scheme 5), 49 (Q1.4) and 49 (Q1.5) (Fig. 22) are given in the text. References to the relevant sources are also there, however, unfortunately, not opposite the corresponding values. We have fixed this annoying mistake.

  1. For the convenience of readers' understanding, it is better to add the notations δH and δC in Table 2.

The notations δH and δC were added in Table 1

  1. As many diastereomers were found and synthesized, were there reported absolute configurations of any compounds?

The absolute configurations of these compounds were not reported

  1. The paragraph in the subsection Conclusions was the same as part of the first paragraph in the subsection Introduction. Please rewrite it. You can list the correct structures of some compounds in this part.

The subsection Conclusions has been rewritten

  1. Please update the information for Reference [58].

The reference [58] has been updated as follows (volume number is added)

 Dragan, S. V.; Borisova, K. L.; Pelageev, D. N.; Anufriev, V. Ph. Concerning the stereoselectivity of the oxidative dimerization of 3‑alkyl-2-hydroxy-1,4-naphthoquinones in the synthesis of hybocarpone. Nat. Prod. Commun. 2019, 14, DOI: 10.1177/1934578X19860687

Comments on the Quality of English Language

The errors indicated by the reviewer have been corrected.